# Identification of Neural Mechanisms in First Single-Sweep Analysis in oVEMPs and Novel Normative Data

**DOI:** 10.3390/jcm11237124

**Published:** 2022-11-30

**Authors:** Dietmar J. Hecker, Hans Scherer, Uwe Schönfeld, Laura Jerono, Armand Koch, Anna-Katharina Rink, Lisa Schulte-Goebel, Maximilian Linxweiler, Mathias Fousse, Alessandro Bozzato, Bernhard Schick

**Affiliations:** 1Department for Otorhinolaryngology, Head and Neck Surgery, Saarland University Medical Center, 66421 Homburg, Germany; 2Charité University Medicine, 13353 Berlin, Germany; 3Department of Otorhinolaryngology, Head and Neck Surgery, Charité University Medicine, 13353 Berlin, Germany; 4Department of Neurology, Saarland University Medical Center, 66421 Homburg, Germany

**Keywords:** oVEMPs, bone-conducted, n10 amplitude and latency, single sweeps, synchronization, novel algorithms, efferent mechanism

## Abstract

Background: Bone-conducted (BC) VEMPs provide important tools for measuring otolith function. However, two major drawbacks of this method are encountered in clinical practice—small n10 amplitude and averaging technique. In this study, we present the results of a new VEMP setup measuring technique combined with a novel single-sweep analysis. Methods: The study included BC oVEMP data from 92 participants for the evaluation of normative data using a novel analysis technique. For evaluating test-retest reliability, the intraclass correlation coefficient (ICC) was used. Results: We found significant n10 amplitude differences in single-sweep analyses after the first and second measurements. Thereby, mathematical analyses of the head movement did not show any differences in the first or second measurements. The normative n10 amplitude was 20.66 µV with an asymmetric ratio (AR) of 7%. The new value of late shift difference (LSD) was 0.01 ms. The test retest-reliability showed good to excellent ICC results in 9 out of 10 measurements. Conclusions: Our results support a phenomenon in single-sweep analysis of the first stimuli independent of head movement and signal morphology. Furthermore, the values obtained with the new measurement method appear to be more sensitive and may allow an extended diagnostic range due to the new parameter LSD.

## 1. Introduction

Clinical testing of otolith function is of high interest during the detailed analysis of the complete peripheral vestibular organ [1]. The utricular function has been suggested to be analyzed by oVEMP measurements [2,3]. Here, the n10-amplitude serves as an indicator of the utricular function. Based on the hearing measurement platform, many manufacturers use acoustic stimuli (tone bursts or click stimuli) in their VEMP measurement devices to activate the vestibulo-ocular reflex. In healthy subjects, the optimum frequency for evoking VEMPs to short-tone burst stimuli is around 500 Hz [1]. Deepak et al. [4] demonstrated that 500 Hz stimuli have significantly better n1-p1 amplitude in oVEMPs as click stimuli. Depending on the pathology, the optimum stimulus frequencies for VEMPs could be different [5].

The oVEMP responses show a very weak signal [6]; averaging techniques typically require 100–200 single sweeps to improve the signal-to-noise ratio (SNR) (details from https://madsen.hu/pdf/tajekoztato/Chartr_EP_200_Insights_November_2008_STD.pdf, accesssed on 10 October 2022). High sound pressure levels in VEMP measurements are necessary to activate the primary vestibular afferent neurons by producing a small movement of the macula. The induced movement by the stapes releases a one-directional shock wave in the fluid of the scala vestibuli [1]. In the case of conductive hearing loss [7] or after cochlea implantation surgery [8], it is however almost impossible to evoke VEMPs by acoustic stimuli.

In contrast to air-conducted sound (ACS), the bone-conducted vibration (BCV) activates the afferent neurons in all directions [1] and these stimuli are suitable for patients with conductive hearing loss and after cochlea implantation. As described above, the evaluation of n10 amplitudes and n10 latency is based on the averaging of 100 to 200 sweeps. As a criterion of dysfunction, the asymmetry ratio (AR) of the n10 amplitude is used. An AR in oVEMPs of >0.4 has been suggested to indicate utricular pathology [9] in vestibular testing. In many studies, VEMPs are used to test the otolith organs. In addition, some studies use VEMPs to diagnose neuronal diseases [10,11]. The n10 latency is delayed in neurological diseases such as multiple sclerosis. Due to chronic inflammation and demyelinating processes of nervous structures signal transmission may be compromised, as observed by oVEMP measuring. Even very small lesions may be detected that were not noticeable in clinical examination or even in MRI scans [10].

In order to establish a new diagnostic method, further verification appears mandatory. Accordingly, test and retest reliability studies with cVEMPs and ACS are mentioned [12,13,14,15], but there is only one publication with test and retest reliability in oVEMPs through head taps stimulation [16]. In this publication, Nguyen et al. found that heap taps stimulation had the highest n10 amplitude as compared to ACS stimulation.

Currently, signal processing of VEMPs is based on averaging single sweeps and comparison of mean amplitudes and latencies. However, information about contamination of the VEMP signal by noise (=spontaneous activities) and/or phase jitter is lost during the averaging process [17]. For this reason, small alterations in the individual characteristic patterns may be difficult to detect. In our recent publications [17,18], we were able to show how insensitively the averaging technique reacts to considerable spontaneous activities and phase jitter in the single-sweeps. In combination with new triggering, our new technique displayed n10 amplitudes over 20 µV with a low asymmetric ratio (AR) in a group of ten healthy volunteers. As few as six head taps were sufficient to achieve a signal stability of >95% in oVEMPs measurements. So far, no further evaluation of this technique has been presented in the literature (statement of Prof. Curthoys in [19]) and we described a formula of phase synchronization vector (PSV) using single-sweep analyses with complex arithmetic (for more details see [18]). The difference between the PSV from the right to the left sides is defined as the delta shift-vector (DSV). In our previous work, it was revealed that this method appears to be sensitive to spontaneous activities and phase jitter of evoked EMG potentials [18].

Here we describe our experiences and observations by testing the new VEMP setup in 92 healthy volunteers. In our single sweep observation, we identified different neural mechanisms and we calculated normative oVEMP data in latencies and amplitudes with a formula of novel features.

## 2. Materials and Methods

### 2.1. Subjects

A total of 92 healthy subjects (mean age of 25 years ± 4 years, ranging from 18 to 42 years, 37 males and 55 females) without any history of hearing or vestibular disorders were investigated for normative data. In addition to the known AR, we have determined a new assessment value for the lateral latency shift based on Li et al. [20]. We define the latency shift difference (LSD) as the latency difference of n10 amplitude from the left side to the right side. All subjects were measured with a minimum of 20 single sweeps per oVEMP measurement.

A total of 64 subjects (mean age of 28 years ± 6.1 years, ranging from 18 to 40 years, 26 males and 38 females) as a subcohort of the main group were used for calculation of the test-retest-reliability in one session (within-session reliability). The identification of novel mechanisms in single-sweep analysis based on a Matlab (Mathworks, Natick, MA, USA) script file (programming by D.J.H.) for automated analysis of differences in mean n10 amplitude of single sweeps.

For analyzing the head movement during head taps stimuli on Fz we used a triaxial linear acceleration sensor (ADXL335, Analog Devices, Norwood, MA, USA) that was fixed on the head (for more technical details see [21]). We recorded synchronously the movements of the head during the head tap stimulation and VEMP measurement in three volunteers.

A total of 25 subjects (mean age 24 years ± 2.5 years, range from 21 to 30 years, 14 males and 11 females) from the whole group were investigated for repeated measurement in two sessions (between-session reliability), with a mean time span of 15 weeks—ranging from 4 to 56 weeks after the first session.

### 2.2. oVEMP—Measurements

We analyzed oVEMPs as previously described by Hecker et al. [18]. Hereby, the positioning of the patients and placement of the surface electrodes were performed as described previously by Nguyen et al. [16] as well as Iwasaki et al. [9].

### 2.3. Statistical Analysis

SPSS version 16 (IBM, Ehningen, Germany) was used for statistical analysis using a significance level of 5% and a statistical power of 80%. In testing the significance of thresholds, the existence of normal distribution was controlled by Kolmogorov-Smirnov testing and homogenous variance was checked by the Levine test. If one of the parameters was identified to be inappropriate, the nonparametric Mann-Whitney-U testing was applied. For calculating the degree of linear dependence of function, we used Pearson’s correlation coefficient. To analyze test-retest reliability, the interclass correlations (ICC) were used, analogous to Nguyen et al. 2010 [16] and we defined a poor reliability for all ICC values < 0.4, a fair to good reliability for all 0.4 < ICC < 0.75, and an excellent reliability for ICC > 0.75.

## 3. Results

### 3.1. Findings of Novel Neural Mechanisms

Figure 1 shows two oVEMP measurements (first left column, second right column) in one session with single-sweeps expressions in Figure 1a,b in healthy humans. In these Figure 1a,b, the amplitude is scaled by color with a visible n10 trace at 10 ms. The middle row in Figure 1c,d shows the meaning of all single sweeps as a solid line and the meaning of the first six single sweeps as a dotted line. The correlation coefficient between these meanings is 97% (left = first measurement) and 98% (right = second measurement), respectively. The bottom line (Figure 1e,f) shows a section of the n10 amplitudes in the averaged signals. Visible are the amplitude differences between the averaging of all single sweeps compared to the averaging of the first six single sweeps. This difference is 2.67 µV in the first measurement (left side) compared to 0.91 µV in the second measurement (right side). These single-sweep illustrations also gave us the impression of lower first signals followed by stable higher signals.

Based on this observation, we checked the n10 amplitude differences in thedependence of head movement on three healthy volunteers. To measure possible changes in head movements due to the stimulation on Fz, a triaxial linear acceleration sensor (ADXL335, Analog Devices, Norwood, MA, USA) was fixed at the head (for more technical details see Iwasaki et al. 2008b [21]). The analysis of the head movements showed no significant differences between the first six stimuli and the following stimuli in three volunteers. The amplitude of the linear accelerations at the head was found to be 0.73 ± 0.062 g (sweeps 1 to 6) in contrast to 0.76 ± 0.067 g (sweeps 7 to 20). The Mann-Whitney-U testing showed no significant difference (*p* < 0.34).

Based on the automated detection (programmed with Matlab 2020, Mathworks, Natick, MA, USA by D.J.H.) we analyzed in the whole group of 64 volunteers (within-session reliability group) the difference in n10 amplitude, based on the algorithm as described before. The following Table 1 shows all calculated values of the n10 amplitude divided in the first (Figure 2a) and the second (Figure 2b) measurements.

Only the distributions of difference amplitude (first vs. the second measurement) had significant values (*p* ≤ 0.038) using Mann-Whitney-U testing (with a significant result in Kolmogorov-Smirnov-test). The following Figure 2 represents the boxplot of the two distributions of difference in n10 amplitude. The analysis of n10 latencies showed no significant differences.

### 3.2. Normative Data of the BC oVEMPs

On average, the n10 amplitudes were found to be 20.6 ± 8.66 µV (median 21.09 µV) on the measured right side and 20.72 ± 9.2 µV (median 21.41 µV) on the measured left side in a total of 92 subjects with no significant difference between the right and left sides (*p* < 0.678 two-tailed *t*-test). The corresponding asymmetry ratio (AR) was 7 ± 5% (see Figure 3).

The average latencies were 10.68 ± 0.84 ms (median 10.73 ms) on the measured right side and 10.66 ± 0.82 ms (median 10.73 ms) on the measured left side, with no significant difference between the right and left sides (*p* < 0.92 two-tailed *t*-test). The corresponding latency shift difference (LSD) was 0.02 ± 0.43 ms (median 0.0 ms), indicating no relevant pathology (see Figure 4).

Figure 5 shows the novel values of PSV (a–c) and DSV (d). These novel parameters result from the complex wavelet transformation of single sweeps, they are normalized and dimensionless. Values on the right measured side were 19.88 ± 3.86 (median 19.03) and 19.81 ± 3.87 (median 18.80 µV) on the left measured side with no significant difference between the right and left side (*p* = 0.57 two-tailed *t*-test). The corresponding DSV was 4.11 ± 3.81 (median 3.73; see Figure 5d), and Table 2 shows the normative data (*n* = 184) with a confidence interval of all volunteers.

### 3.3. Test and Retest Reliability of BC oVEMPs

In the next step, we examined the oVEMP signals on reliability in session and time. Based on 128 measurements (64 volunteers) we calculated the test and retest reliability in one session. Table 3 shows the amplitudes, latencies, and AR of 128 data sets with mean ± standard deviation (std), median, and ICC. The dimensions in Table 3 and Table 4 are µV for amplitudes and ms for latencies. The ICC value and the AR are dimensionless.

In all distributions, no significant difference could be detected in the *t*-test. The ICC showed in three cases excellent (n10 amplitude, n10 latency left) and in two cases fair to good (n10 latency right, AR) values. No distribution had poor reliability.

In contrast to the aforementioned measurement (one session), we calculated the test-retest reliability of oVEMPs in a group of 25 subjects in two sessions (about 15 weeks after the first session).

In all distributions, no significant difference was detected in the t-test. The ICC showed in two cases excellent (n10 amplitude right and left), in two cases fair to good (n10 latency right and left) values, and one distribution (AR) showed poor reliability.

## 4. Discussion

It is established in clinical practice that BC oVEMP measurements represent a useful additional technique in functional vestibular assessment [22]. In combination with the powerful “Minishaker” 4810 and the power amplifier 2718 (both Brüel & Kjaer, Naerum, Denmark) it is possible to use the ICS Chartr EP 200, otometrics (Natus Medical Incorporated, 3150 Pleasant View Road, Middleton, WI, USA) for BC VEMP measurements. One disadvantage of this technique is the fact that each side must be measured separately, although VEMPs can be measured on both sides in one session [1,23]. As described above, the evaluation of n10 amplitudes and n10 latency is based on the averaging of 100 to 200 sweeps. Here, we describe the results of a new VEMP measurement setup in combination with single-sweep analysis. In our study [18] we revealed that very stable VEMPs signals (over 95%) were reached after a smaller number of head taps (oVEMPs < 6, cVEMPs < 11) compared to current recommendations. Electrically evoked responses are extremely susceptible to disruption by electrical noise and stimuli interferences (e.q. stimulus artifact). Our setup method is completely free from stimuli interference.

### 4.1. Findings of Novel Neural Mechanisms

Based on single-sweep analysis, we demonstrated that there exist significant differences in the first single sweeps between the first and the second measurements in one session. Analysis of head movement with a triaxial linear accelerator during head taps stimulation did not show any differences. Furthermore, the mathematical analysis via Pearson’s correlation coefficient showed a perfect match of signal morphologies. We assume that the significant amplitude difference is due to differences in neural processing. We speculate that a possible relationship between the hair cells of the cochlea and the hair cells of the equilibrium organ may be a biological correlate for this observation. It is known that some animals are able to compensate for enormous head impacts without trauma to the vestibular organ (e.g., woodpeckers, musk ox, etc. [24,25]). To date, it has not been possible to clarify how these animals protect their vestibular organs from these mechanical stresses [26]. From the cochlea, it is known that we can control the sensitivity of the outer hair cells (OHC) via efferent mechanisms [27]. We hypothesize that there are exit hair cells in the organ of equilibrium that try to prevent damage to the head by a reflex mechanism.

There are 275 million soccer players worldwide (https://soccerprime.com/how-many-soccer-players-in-the-world/, accessed on 10 October 2022), and soccer is the only contact sport with purposeful use of the head for controlling and advancing the trajectory of the ball. Head contact in soccer has the potential to cause traumatic head injury. Many studies show that most head injuries result from contact with other players in the heading duels.

While observing slow-motion recordings of animals banging their heads against each other or woodpeckers hitting tree trunks with their beaks, you can see that they all close their eyes just before the impact [28]. We also observe the same phenomenon in soccer players who consciously execute a header [29]. Although high forces act on the head in such situations, neither the animal nor the soccer player has head injuries. Interestingly, Boden et al. [30] found that none of the concussions resulted from the intentional heading of the ball in their study. After unintentional head contact in soccer, players often still suffer from dizziness. Accordingly, closing the eyes seems to trigger a protective mechanism.

For cVEMPs [13,31] the authors compared the p13n23 amplitudes with and without feedback method and found greater p13n23 amplitudes in sessions without feedback.

Brook et al. 2014 [32] repeatedly measured tVEMPs (for more details see [33]) depending on the patient awareness. In the first session, two measurements were performed with the volunteers’ eyes open during stimulation. In the second session, two measurements were performed with the volunteers’ eyes closed during stimulation.

In the first session (volunteers’ eyes were open during stimulation), the p13n23 amplitude was lower during the first measurement than in the second measurement. With eyes closed, the results were the inverse. It remains unclear where these differences originate from.

Nguyen et al. (2010) [16] described the n10 amplitude of 4.16 ± 3.45 µV in the first session and 2.71 ± 2.53 µV in the second measure with clicks. In our opinion, the lower amplitude could also have led to a blockage of the vestibular organ due to strongly increased stimulus intensity (up to 140 dB peak SPL). Only in this way can we explain why there are differences in the amplitudes.

Based on our results, we strongly encourage further studies to investigate the efferent influence on the organ of equilibrium. The function of the efferent vestibular nerves has been described in several publications [34,35,36]. Two possible efferent mechanisms can be considered: a direct pathway from hair cells type II, or/and an indirect pathway to the peripheral segment over the afferent vestibular nerve, either near type I hair cells or to its synapse very close to the hair cell’s body. For rapid inhibition, only the direct projection to the hair cell type II appears logical.

### 4.2. Normative Data of the BC oVEMPs

In addition to the conventional parameters of mean, standard deviation, and median, it was also important for us to calculate and indicate confidence intervals. We also compared the right and left sides. Both the confidence interval and the side comparison in the normative data have not been described before. The observation of clearly detectable high n10 amplitudes (20.6 ± 8.9 µV vs. 7.44 ± 3.78 µV [9] or 6.38 ± 4.21 µV [16] with an excellent/good test-retest stability in the presented study underlines the suggestion to integrate oVEMP-measurements in the clinical setting to analyze the utriculus function. As a criterion for dysfunction, the asymmetry ratio (AR) of the n10 amplitude is used. An AR in n10 amplitude of >0.4 has been suggested to indicate utricular pathology [9,21] in vestibular testing. The AR in our study was calculated at 7 ± 5%. The AR of n10 amplitude, applied with head taps in Nguyen et al. [16], was in the first 38.93 ± 27.38% and in the second session 41.33 ± 23.11%.

As the asymmetry ratio (AR) has been used as a good indicator for the symmetry of the utriculus function [1] and evaluations of latencies and amplitudes are commonly used in auditory evoked potentials [37], we defined the latency shift difference (LSD) as a novel indicator for symmetry/asymmetry neural myogenic retrolabyrinthine response: LSD = (n10 latency right measured—n10 latency left measured). A highly positive/negative LSD value indicates that the left/right utriculus has a retrolabyrinthine deficit.

In our study, we described and calculated the new parameter LSD with 0.01 ± 0.39 ms as a potential novel indicator for neurological diseases. The combination of traditional (amplitude, latencies) and novel parameters (LSD, PSV, DSV) has the potential to enable a better differentiation between the inner ear and retrolabyrinthine deficits [17].

### 4.3. Test and Retest-Reliability of BC oVEMPs

To prove the test and retest reliability, we performed two similar measurements in (i) one session (*n* = 64) and (ii) two measurements at intervals of at least four weeks (*n* = 25). To date, test and retest reliability studies with cVEMPs and ACS exist [12,13,14,15], but there is only one publication with test and retest reliability in oVEMPs with head taps stimulation [16]. Our time difference in (ii) was in line with this publication. In our data analyses, we could show differences depending on the measurement side. Although these differences are not significant, we performed the reliability test separately for each side.

When we compared the ICC values of Nguyen et al. [16] with our data, we obtained better results in all parameters for (i). In (ii), except for the AR, all other ICC values are better or at the same level. The low ICC value for the AR is due to the ratio between the mean value and the standard deviation. In our data we have 0.08 ± 0.06% and 0.09 ± 0.07%, compared to 38.93 ± 27.38% and 41.33 ± 23.11% in Nguyen et al. [16].

In our study, we analyzed BC oVEMPs with a novel algorithm for the analysis of single sweeps based on datasets from 92 healthy subjects, and we described normative and novel data with confidence intervals. Those results are highly relevant for clinical practice. Furthermore, we observed significant differences in the first single sweeps compared to the second measurement in one session. In light of the present findings, we suspect that efferent innervations may be held responsible for posing as a protective mechanism of the human vestibular sensory system.

## 5. Conclusions

In our study, we analyzed BC oVEMPs with novel triggering and a novel algorithm for the analysis of single sweeps based on datasets from 92 healthy subjects. We observed significant differences in the first single sweeps compared to the second measurement in one session, which disappeared in the further course. In light of the present findings, we suspect that efferent innervations may be held responsible for posing as a protective mechanism of the human vestibular sensory system.

The common normative data in our study illustrates the potential of those newly established techniques. In combination with the novel parameters LSD, PSS, and DSV, we believe that this technique is a promising tool for advancing clinical diagnostics by distinguishing between labyrinthine and retrolabyrinthine deficits [17].

## Figures and Tables

**Figure 1 jcm-11-07124-f001:**
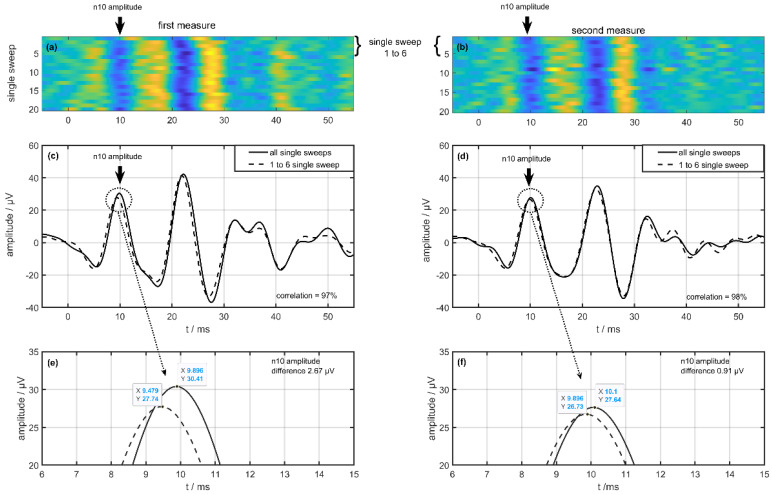
Difference from one volunteer on a single-sweep algorithm between the first (left column) and second measurements (right column) during one examination. (**a**,**b**) show the single sweeps with a color map on the amplitude and (**c**,**d**) show the result of averaging the single sweeps (dotted line 1 to 6 single sweep, solid line all single sweeps). The bottom row (**e**,**f**) shows a view from the n10 amplitude of the averaged signal.

**Figure 2 jcm-11-07124-f002:**
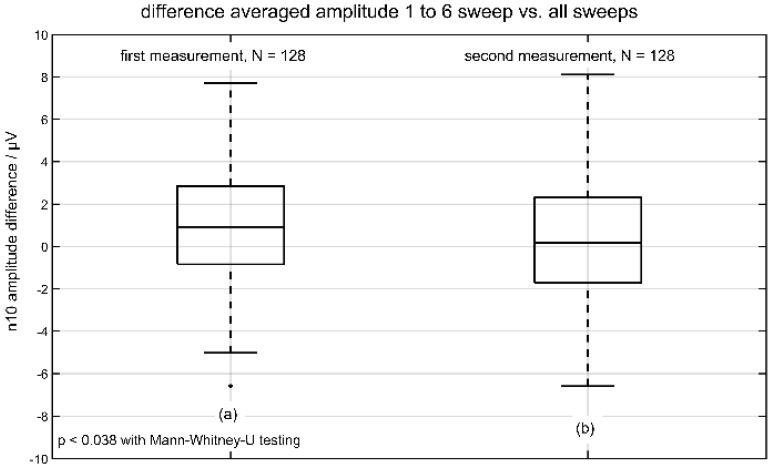
Differences in averaged n10 amplitude (sweeps 1 to 6 vs. all sweeps). The result of the first measurement is depicted in (**a**) and the result of the second measurement is shown in (**b**).

**Figure 3 jcm-11-07124-f003:**
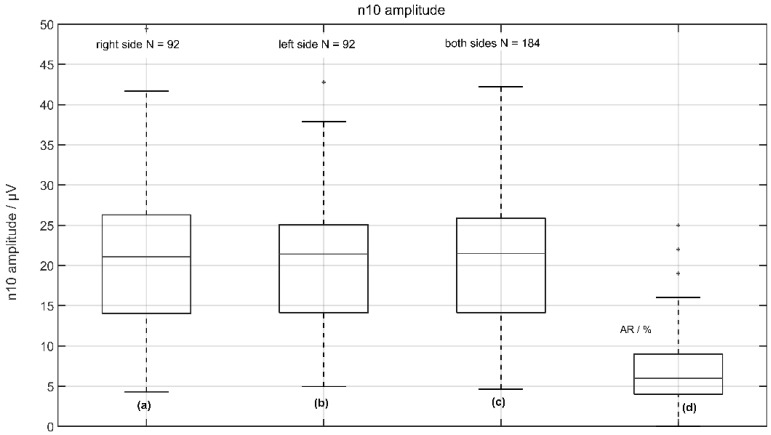
Normative data of n10 amplitude from 92 volunteers: (**a**) the response of the left side (right measured); (**b**) the response of the right side (left measured); (**c**) the response from left and right sides; and (**d**) the asymmetric ratio (AR) of n10 amplitudes.

**Figure 4 jcm-11-07124-f004:**
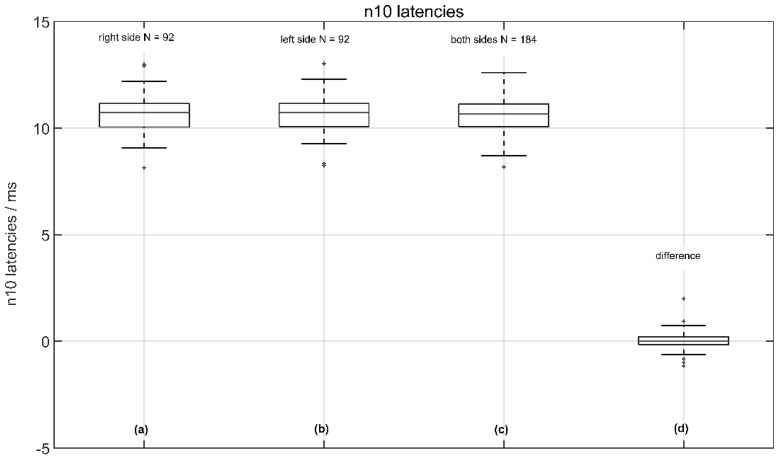
Normative data of n10 amplitude latencies from 92 volunteers: (**a**) the response of the left side (right measured); (**b**) the response of the right side (left measured); (**c**) the response from the left and right sides; and (**d**) the (latency shift difference (LSD): right side—left side).

**Figure 5 jcm-11-07124-f005:**
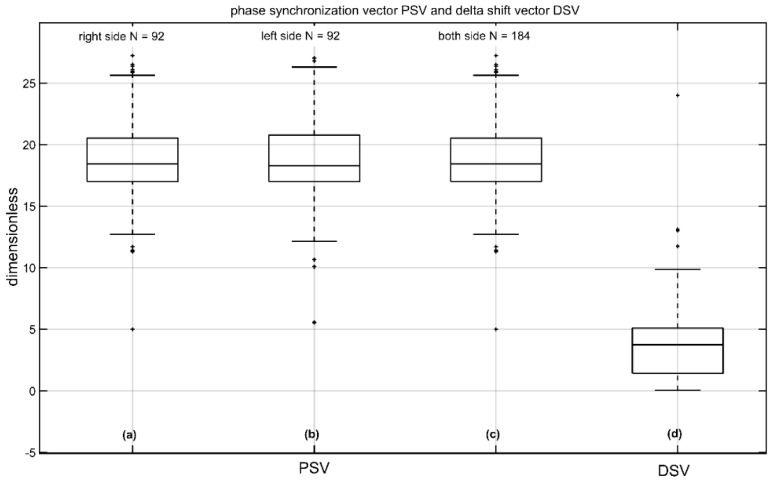
Phase synchronization vector (PSV) and delta shift-vector (DSV) of n10 amplitude from 92 volunteers; (**a**) the PSV of the left side (right measured); (**b**) the PSV of the right side (left measured); (**c**) the PSV from left and right side; and (**d**) the DSV (difference: right PSV—left PSV). The unit of amplitude is µV and the unit of the latencies with LSD is ms. The parameters AR, PSV, and DSV are normalized and dimensionless.

**Table 1 jcm-11-07124-t001:** Calculated values of the n10 amplitude with mean, median, and standard deviation (std) for the first and the second measurements. ***** The difference of n10 amplitude (the first vs. the second measurements) was significant (*p* ≤ 0.038) in the Mann-Whitney-U testing.

	First Measurement *n* = 128			Second Measurement *n* = 128		
	Mean	Median	Std	Mean	Median	Std
n10 amplitude from all sweeps	18.8	17.2	9.6	18.9	17.6	9.8
n10 amplitude from 1 to 6 sweeps	17.7	16.0	9.5	18.5	16.9	9.5
Difference *	1.1	0.92	2.6	0.4	0.1	3.9

**Table 2 jcm-11-07124-t002:** Normative data (mean, median, and standard deviation (std)) with a confidence interval of 92 volunteers divided into 184 measurements of n10 amplitude, n10 latency, and asymmetric ratio (AR) with novel parameters of phase synchronization vector (PSV), delta shift-vector (DSV), and latency shift difference (LSD).

				Confidence Interval	
*n* = 184	Mean	Median	Std	Lower	Upper
n10 amplitude	20.6	21.2	8.9	19.4	22.0
n10 latency	10.6	10.7	0.8	10.5	10.8
AR	0.07	0.06	0.048	0.05	0.08
PSV	19.8	19.0	3.9	19.1	20.6
DSV	4.1	3.7	3.8	3.1	5.1
LSD	0.02	0.0	0.43	−0.71	+0.11

**Table 3 jcm-11-07124-t003:** Calculated amplitudes and latencies (side-specific) of the n10 wave and asymmetric ratio with mean ± standard derivation (std), median, and interclass correlation coefficient (ICC) depending on first and second measurements in one session.

One Session	n10 Amp. Right		n10 Amp. Left		n10 Latency Right		n10 Latency Left		n10 AR	
*n* = 64	First	Second	First	Second	First	Second	First	Second	First	Second
Mean ± std	18.9 ± 9.27	18.8 ± 9.1	18.6 ± 9.8	18.9 ± 10.3	10.5 ± 0.71	10.5 ± 0.78	10.6 ± 0.73	10.6 ± 0.88	0.11 ± 0.09	0.14 ± 0.11
Median	17.67	18.0	16.2	16.5	10.5	10.4	10.7	10.6	0.09	0.13
ICC	0.91		0.93		0.72		0.76		0.65	

**Table 4 jcm-11-07124-t004:** Calculated amplitudes and latencies (side-specific) of the n10 wave with median, mean ± std, and interclass correlation coefficient (ICC) depending on the first and the second measurements in two sessions.

Two Sessions	n10 Amp. Right		n10 Amp. Left		n10 Latency Right		n10 Latency Left		n10 AR	
*n* = 25	First	Second	First	Second	First	Second	First	Second	First	Second
Mean ± std	18.8 ± 6.64	18.2 ± 5.68	19.2 ± 7.05	19.2 ± 6.87	10.7 ± 0.6	10.6 ± 0.77	10.8 ± 0.7	10.64 ± 0.70	0.08 ± 0.06	0.09 ± 0.07
Median	19.6	18,9	19.6	19.9	10.8	10.6	10.9	10.7	0.07	0.09
ICC	0.88		0.84		0.67		0.68		0.20	

## Data Availability

The data presented in this study are available on request from the corresponding author.

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
