# Peer review of "Identification of Neural Mechanisms in First Single-Sweep Analysis in oVEMPs and Novel Normative Data"

_jcm, 2022, doi:10.3390/jcm11237124_

Round 1

Reviewer 1 Report

Study is good and good attempt is done to collect good data. However, following need to be addresed.

Some errors

124 : to check on significance level. Is it referred to p value. If so value should be different.

137: Fig 1a and 1b are referred but in the figure 1 their labeling is missing.

136: Good to Figure 1a, 1b instead of left and second right. If required to give more sub-lables.

291. paragraph format error

301: spell error “feedback”

306: “grater” should be “greater”

 Some suggestions to make it better

1.  There are good number of studies done by Dr. Niraj Kumar Singh and Dr. Sujeet Kumar Sinha in this area. It may be good to review and add them.

2. Evoking of responses due to BC stimulus, contamination of the evoked responses and reliability of the results(especially when ref. to ear specific) needs to be explained/ strengthen better in the discussion. 

3. Take home message which includes protocol, data interpretation can be highlighted.

4. Authors have used matlab code to autodetect the responses. Do they recommend this to be done as routine clinical practice. If so, good to share to share the codes. If do not recommend, what is the advice given based on the outcome of the study to identify responses.

5. I am not competent on english, however, noticed some spell errors. Good to have review on that. 

Author Response

dear reviewer 1:

thank you for your revision and your good advice.
I would like to make the following comments.

1.) Errors have been corrected, the manuscript has been completely proofread again. 

2.) Publication by Kumar have been taken into account , thanks again for the good advice :-)

3.) Further notes on the contamination of evoked potentials have been incorporated

4.) Take home message and data interpretation were supplemented

5.) matlab code: I wrote the VEMP APP (attached VEMP APP.JPG) myself and we use it for diagnostics. Technical details are described in the IEEE publication Hecker et. al 2014. The analysis software requires single sweeps. Worldwide, only we are able to collect these.

Reviewer 2 Report

1. What is the main question addressed by the research?

The authors studied the neural mechanisms in first single sweep analysis in oVEMPs

2. Do you consider the topic original or relevant in the field? Does it
address a specific gap in the field?

The study proposal of this topic is very interesting.

3. What does it add to the subject area compared with other published
material?

The conclusion of the study is promising and could be an advance in the diagnosis of patients with vestibular pathology.

4. What specific improvements should the authors confider regarding the
methodology? What further controls should be considered?

Enlarging the study sample can confirm and make the data more robust.

5. Are the conclusions consistent with the evidence and arguments
presented and do they address the main question posed?

The conclusions agree with the investigation carried out and the results obtained.

6. Are the references appropriate?

References are adequate and up-to-date.

7. Please include any additional comments on the tables and figures.

The figures and images are well organized and didactic for a better understanding of the text.

Author Response

Dear reviewer
Thank you very much for your review.